# Advances in Precision Health and Emerging Diagnostics for Women

**DOI:** 10.3390/jcm8101525

**Published:** 2019-09-23

**Authors:** Megan B. Fitzpatrick, Avnesh S. Thakor

**Affiliations:** 1Interventional Regenerative Medicine and Imaging Laboratory, Department of Radiology, Stanford University, Palo Alto, CA 94305, USA; megan6@stanford.edu; 2Stanford Hospitals and Clinics, Department of Pathology, Stanford, CA 94305, USA

**Keywords:** precision health, women’s health, Femtech, MedTech, preventive health

## Abstract

During the Dutch winter famine of 1944–1945, an interesting observation was made about the offspring born during this time—They had an increased risk of developing metabolic syndrome and other chronic diseases. Subsequent research has confirmed this finding as well as noting that health outcomes for many diseases are different, and often worse, for women. These findings, combined with the lack of enrollment of women in clinical trials and/or analysis of sex-specific differences are important factors which need to be addressed. In fact, Women’s health research and sex differences have historically been overlooked or lumped together and assumed equivalent to those of men. Hence, a focus on women’s health and disease prevention is critical to improve the lives of women in the 21st Century. In this review, we point out the critical differences biologically and socially that present both challenges and opportunities for development of novel platforms for precision health. The technologic and scientific advances specific to women’s precision health have the potential to improve the health and wellbeing for all females across the world.

## 1. Introduction

Women’s health, when in balance, is cyclical. In Celtic lore, the Triqueta (Trinity knot) represents three unique phases of a woman’s life: Maiden, Mother, Crone; and each stage presents unique health changes. Fundamental biologic sex differences influence even the microscopic composition of the female body [1,2,3]. These sex differences are further influenced by environmental and nutritional exposures during fetal development. Indeed, observations from the offspring of Dutch women exposed to wartime famine (1944–1945) revealed that these individuals were at an increased risk for developing metabolic syndrome in later life. Consequently, a Dutch researcher (Dr. Barker) postulated that in utero exposures influence lifelong health outcomes which subsequently became known as the Barker hypothesis [4,5]. Further research has linked environmental exposures in early development with a range of adverse health consequences including cardiovascular disease, autoimmune diseases, infertility, cancers, and more. The effect of these exposures has also been shown to be more profound amongst women [4,5].

Despite advances in medical technology and health care, the death rate among middle-aged American women is increasing [6]. The differing biology of women alters both their risk for, as well as their presentation of, diseases. Furthermore, childbearing and the diseases of the female genital tract present unique health challenges not encountered by men. Optimal health throughout all the stages of a female’s life can prevent disease. In this way, *precision health* offers the promise to provide personalized, female-specific lifestyle coaching, personal health education, and early disease detection to mitigate the consequences of diseases throughout a woman’s life. This contrasts with *precision medicine* which focuses on molecular targeting to diagnose and treat diseases thereby affecting the outcome after the onset of disease.

Novel devices and mobile applications are being developed specifically for women to optimize their health and provide access to information for prevention of disease. Indeed, it is predicted that health technologies for women will likely comprise a $50 billion industry by 2025. These personalized diagnostics, smart-tools, and evidence-based recommendations have the propensity to change how we understand and address women’s health with a focus on well-being and prevention. In the emerging “FemTech” market, diagnostics that utilize various body fluids (i.e., saliva, urine and breast milk) can now detect diseases (i.e., cancers, endometriosis) as well as monitor hormone levels and even nutritional status (Figure 1). In this review, we will explore emerging technologies which can optimize the health of females through their lifecycle. Female Lifecycles can be seen in Figure 2.

## 2. Maiden

### 2.1. Puberty

Young women develop many of the health habits during puberty that last for their lifetimes. The first signs of female puberty typically start around age 9 with a growth spurt and increased fat deposition. Shortly thereafter, breast development begins accompanied by maturation and growth of reproductive organs. First menstruation (menarche) and ovulation are the most definitive signs of puberty in women. In the Northern Hemisphere, 95% of females reach menarche between 11 to 15 years of age [7]. Pubertal changes also bring unique challenges to this age group, particularly sexual and menstrual health (see below), as well as mental health and physical development. Educational and interactive technologies are now emerging to support young women through these unique phases.

### 2.2. Menstrual Health

The typical menstrual cycle is on average 28 days (21–31 days in adults; 21–45 days in adolescents). The cycle is divided into four phases: follicular, proliferative phase, ovulatory and secretory phases. Luteinizing hormone (LH) and estrogen (E_2_) both surge just prior to ovulation, to stimulate the release of an egg/oocyte from a dominant follicle and this occurs around day 14 of an average 28 day cycle [7]. If fertilization and implantation of the oocyte does not occur, progesterone increases as the secretory phase ensues; this prepares the endometrium for shedding during menstruation. Accurate measurement and logging of symptoms during menstruation can assist in identifying potential causes of cycle irregularity, cycle-related health problems and issues with contraception [8]. Some refer to menstrual cycle health as a vital sign among women since it can offer insight into a range of other health problems. The ability to track the menstrual cycles and share this information with healthcare providers may provide deeper insight into a woman’s health [9,10].

Technologies focused on pubescent girls include the Puberty Clues mobile application, which help teens to learn about changes in their bodies and emotional lives in a private, personalized, and interactive way. The Lola First Period Kit offers products and tracking support to help teens understand their early menstrual cycles. Tracking cycles is often accompanied by tracking premenstrual syndrome symptoms. These features could potentially help women and their health providers understand their cycles, as well as detect warning signs of other conditions such as polycystic ovarian syndrome, endometriosis, and fibroids.

Beyond puberty, menstrual cycle tracking and pregnancy-avoidance timing are also available through cycle trackers such as the FDA-approved Dot. This algorithm-based mobile application uses a women’s period start date and gathers historical cycle data to predict the chance of pregnancy for each day of the menstrual cycle. As the app “learns” about a woman’s cycle over time, it personalizes her fertile window—The days of her cycle when pregnancy is likely. A study evaluating 14 different mobile applications found that Lady Cycle, mNFP, and Lily, based on the Sensiplan method, could consistently determine the correct fertile window [11,12,13,14,15]. That said, the authors emphasize that for those wanting to achieve or avoid pregnancy, women should also identify the cardinal signs of fertility, such as cervical mucus, to increase the accuracy of tracking. In general, researchers have found that some of these apps had a typical-use failure rate of 5% and a perfect-use failure rate of 1%, which makes these methods comparable to family planning methods such as the pill, vaginal ring, and other fertility awareness-based methods [11,15,16,17]. That said, these applications have decreasing effectiveness with irregular menstrual cycle lengths, inconsistent use, or miscalculation of fertile windows. Although these applications allow for the tracking of symptoms, dates of intercourse, and dates of menstruation, few offer any insight to help identify potential health issues or are used by medical professionals; hence, associating/linking this data in the future may enable medical professionals to identify potential health related issues earlier.

Personalized contraceptive health may be on the horizon as well with a better understanding of pharmacogenetics. For example, a recent study published recently found that female carriers of the cytochrome P3A7*1C variant (adult expression of CYP3A7) have increased metabolism of all steroid hormones [18]. In a third of the women with this variant, Etonogestrel concentrations fell below threshold for ovulatory suppression—Meaning that women were still able to get pregnant despite receiving the contraceptive implant [18]. Hence, oral contraceptive pharmacogenetic testing is now being offered through companies, some in CLIA and CAP certified labs, such as Admera health, Ignite pharmacogenetics, and Genelex. This information therefore has potential implications for the future of contraception recommendations and may help personalize oral contraceptive choice and dosage.

### 2.3. Sexually Transmitted Infections

Despite decades of declining incidence, the rates of syphilis and *Neisseria gonorrhea* are now increasing [19]. Infertility secondary to undiagnosed chlamydia infection also continues to be common despite advances in diagnostic tests (i.e., improved nucleic acid tests); this is likely due to barriers which still exist in getting access to these tests. To address this issue, several self-collection and home testing devices are being developed to enable improved access and usability of sexually transmitted infection (STI) tests. A meta-analysis of studies analyzing the sensitivity and specificity of self-collected samples compared to clinician-collected samples have found adequate analytic agreement with comparable sensitivity and specificity [20]. Emerging data supports that these approaches can improve increase testing rates and identification of people at risk. For example, when mobile technology has been used for the notification of sex partners exposed to STIs (including HIV), more people were notified and screened which led to increased number of new infections being found [21]. The response times, and time to treatment, are also improved with the use of this technology [21]. Employing new diagnostics and linking them with access to the appropriate care and support may therefore hold promise in providing private and accessible screening to patients, as well as partners, who would otherwise be unlikely to be tested.

### 2.4. Other Conditions

Endometriosis is defined as the presence of endometrial glands and stroma in organs other than the native endometrium. Displacement of endometrial tissue cause painful menstrual cycles and can contribute to infertility. Currently, this disease is diagnosed using clinical symptoms and nonspecific findings on ultrasound, pelvic exam, and magnetic resonance imaging; however, there are no features or biomarkers currently available to definitively diagnose endometriosis without histologic confirmation with biopsies obtained via laparoscopic surgery [22]. Emerging non-invasive tests saliva testing for microRNAs (i.e., endogenous noncoding function RNA fragments) have been shown to correlate with endometriosis. This concept has been commercialized in a new company called Dotlab [23].

## 3. Mother

### 3.1. Fertility and Pregnancy

According to data from the Centers for Disease Control and Prevention (CDC), 10% of women (6.1 million) in the US have difficulty getting or staying pregnant [24,25]. With the help of reproductive technology, this rate has dropped slightly between 1982 and 2010 amongst married women, but has slightly increased amongst all women (with any marital status) [25]. Nationwide, women are having children at older ages, which influence these numbers. Given the interest in fertility optimization, several technologies have emerged to provide resources for women to optimize fertility tracking and identify issues with fertility early.

### 3.2. Planning for the Future: Ovarian Reserve and Follicle Tracking

Ovarian reserve testing refers to an indirect estimate of the number and quality of ovarian follicles as an estimate to predict fecundity. As more women have children at older ages, there is an increased demand for predicting their fertile window. Ovarian reserve can be estimated through measurement of serum markers as a proxy for the function of somatic cells within follicles [24,25,26]. One such hormone marker is anti-Mullerian hormone (AMH), which is correlated with primordial ovarian follicles, and declines through a woman’s reproductive cycle [26]. Although this provides the most common “fertility test”, it does not measure the quality of the remaining oocytes [26]. At home finger-stick hormone testing for AMH, FSH, E_2_, LH, thyroid-stimulating hormone, free thyroxine, prolactin, and testosterone is offered by a new company (Modern Fertility) [27]. The laboratory results are then added to a graph overlay with average risk by age in a “fertility timeline”.

### 3.3. The Future: Molecular Testing to Identify Women-Specific Diseases

Tracking the menstrual cycle, especially ovulation, is critical for timing intercourse in order to conceive [11,12,28]. Numerous devices and tracking applications have been developed that exploit different body changes during the menstrual cycle to predict the timing of ovulation and hence “fertile windows” [12,28,29,30,31]. The Symptothermal Method or Fertility Awareness Method (FAM) utilizes the expected increase in body temperature, change in viscosity of cervical fluid, and positioning of the cervix that occur near ovulation. Women who use the FAM method for either contraception or fertility, have variable pregnancy success rates [13,15]. One factor which reduces the effectiveness of this approach is patient compliance. Hence, devices and tests offer improved tracking of changes with or without urine LH levels (which surge 12–24 h prior to ovulation). Although studies have found that these resources can increase the efficacy (correct use) and effectiveness (actual use) of these methods, the amount that it improves compliance is controversial and variable [15,16,28]. For example, smart thermometers that link to the menstrual cycle and symptom tracking apps (e.g., Kindara, Daysy) claim a high rate of accuracy in predicting ovulation, although independent studies on these products have yet to be published, and there are issues with the accuracy of available publications [16,32]. Other tracking devices monitor physiological changes throughout the menstrual cycle to predict ovulation. For example, OvaCue uses the Electrolyte Method™ to track changes in the electrical resistance and ionic concentrations in the saliva and cervical mucus to predict the window 5–7 days prior to peak fertility as well as provide confirmation of ovulation [29,30]. Percept has developed a piezoelectric sensor disc that is placed under the bed mattress which utilizes the predictable physiologic increases in resting heart rate, heart rate variability, and a decrease in breathing rate to allow for ovulation prediction [28]. Wearable devices such as the Ava bracelet similarly track physiologic changes, and initial studies have found 89% accuracy in prediction of ovulation [28,33]. In addition to widely available over the counter urine LH test strips, “smart devices” such as the FDA-approved Mira device can collect serial LH measurements, sync with mobile apps that use artificial intelligence to provide a more sensitive prediction of the LH surge and hence ovulation.

As research continues to understand more on the molecular mechanisms of chronic diseases, companies such as Celmatix are offering genomic DNA tests to link single nucleotide variants to diseases such as endometriosis, polycystic ovarian syndrome, idiopathic infertility, hyperstimulation syndrome, and recurrent pregnancy loss. The implications of detection of single nucleotide variants remain unclear given the contribution of epigenetics and environmental factors involved in disease development [34]. For example, genetic association studies have identified single nucleotide variants associated with endometriosis, but these presumed etiological mutations are neither required for disease development nor detected in all cases of those with disease [22,34].

### 3.4. Pregnancy Testing and Monitoring

Human pregnancy is, on average, 40 weeks of gestation starting from the first day of the last menstrual period. It is generally divided into three trimesters, each lasting about 12–14 weeks, which corresponds with distinct phases of growth and development of the fetus.

#### 3.4.1. Non-invasive genetic testing

Early detection of chromosomal and genetic abnormalities has been part of prenatal screening since the mid-1960s. The most common chromosomal abnormalities that are the target of screening are chromosomal abnormalities in which an extra chromosome is present (trisomy). The primary chromosomal abnormalities targeted by screening are Down syndrome (trisomy 21), Edwards syndrome (trisomy 18), and Patau syndrome (trisomy 13). Prenatal screening initially began amniocentesis, a procedure whereby fluid is collected via a large bore needle inserted into the amniotic sac. Later, a panel of predictive serum markers was developed first targeted at specific high-risk populations for early detection of chromosomal abnormalities. Recently, cell-free DNA testing is an emerging non-invasive alternative to amniocentesis and chorionic villous sampling to detect chromosomal conditions as well as sex chromosomal conditions like Turner and Klinefelter syndrome [34]. Prenatal screening is now expanding to include genetic carrier testing for inheritable diseases, inheritable cancer genetics, and may soon include genes predisposing women to pre-eclampsia [35]. Companies such as Progenity (AscendatMDX) are offering these tests directly to the consumer.

#### 3.4.2. Labor monitoring

Labor commences with uterine contractions to prepare the cervix for delivery of the baby and placenta. There are now devices changing the way pregnancy and delivery are monitored, such as Bellabeat Shell which can amplify the sound of a baby’s heartbeat, and a wearable pregnancy sensor (Bloomlife) that can count and time contractions (either Braxton Hicks or true labor).

### 3.5. Postpartum Care

The post-partum period brings unique challenges which have become the target of emerging technologies, including balancing career and motherhood, breastfeeding, weight and stress management, and urinary incontinence.

#### 3.5.1. Breastfeeding

Breastmilk contains critical nutrients for proper nutrition of the human infant. The nutritional composition of breastmilk is heavily influenced by the nutrition of the mother. For example, docosahexaeonoic acid (DHA; a long-chain omega-3 polyunsaturated fat) is critical for the structure, development and function of the central nervous system as well as for cognitive function, however, its level in breast milk varies based on maternal intake [36]. Likewise, the levels of trace minerals are vitally important for post-natal development of the infant, but they also vary significantly in women from different populations [37]. MyMilk is a new company that offers nutritional analysis of the vital components in breastmilk (e.g., vitamins B6, B12, A, calories, fat) offers nutritional recommendations for breastfeeding moms.

#### 3.5.2. Urinary incontinence

The stress and damage caused during pregnancy and labor can affect the pelvic floor (the supportive muscles between the tailbone and public bone) that and impact daily life activities [38]. Pelvic floor issues are common, affecting up to 70% of women who have recently given birth [38,39]. Elvie is an example of a device created using biofeedback technology to guide pelvic floor exercises for recovery [39,40].

## 4. Crone

### Menopause

Health concerns that present during the peri-menopausal and menopausal transition can significantly impact the quality of life of women as levels of E_2_ fall due to their decreased production by the ovaries. Sleep is often disturbed by hot flashes and the effects of hormone fluctuations. Depression and mood swings often also are common among this age group. Bone density, the cardiovascular system, and breast cancer risk are also influenced by menopause, particularly an early onset menopause. Saliva tests are available to determine hormone levels (particularly estrogen, anti-Mullerian hormone, and follicle stimulating hormone) to monitor perimenopausal hormonal changes (e.g., ZRT laboratories). Prediction of the onset of menopause has also been explored, with increased focus on anti-Mullerian hormone (AMH) levels, although the precision of AMH levels for prediction of menopause remains limited [26,41]. The pharmacogenetic mechanisms involved in estrogen metabolism have also been investigated in relation to menopause and efficacy and metabolism of exogenous estrogen therapy [41,42].

Emerging technologies and mobile applications focused on providing support and insight into the emotional and physical changes of menopause are increasingly available. For example, Menopause View and MyFlo mobile applications provide symptom trackers, educational resources, and peer support through this transition. In addition, Madorra has developed a non-invasive device to treat vaginal dryness. Given the relative lack of available resources and technologies available for women’s needs in this stage, additional focus on the needs specific to peri and post-menopausal women is warranted.

## 5. Topics throughout Women’s Lifecycles

### 5.1. Cardiovascular Disease

Currently, more women die of cardiovascular disease than men [43]. Cardiovascular disease is the number one cause of death among women in the United States [6]. Women manifest cardiovascular disease symptoms differently. In premenopausal women, estradiol (E_2_) has an overall cardio-protective role which helps prevent vascular dysfunction and cardiovascular disease. Due to the influence of E_2_, women’s blood pressure and platelets react differently under stress: buildup of platelets and reduction in microvascular blood flow can cause ischemia in critical organs such as the heart [44]. Therefore, the symptoms and the clinical manifestations of a heart attack are different among pre-menopausal women [43,44,45,46]. When women lose the protection of E_2_ in early menopause, they have a 33% higher risk of coronary heart disease and stroke [47]. The Women’s Ischemia Syndrome Evaluation found that women’s arteries can appear healthy on angiogram, but can have microvascular dysfunction [48,49]. Recent developments in biomarkers for risk stratification and improved diagnostic tests allow for more sensitive detection of cardiovascular disease among women [44,46,49]. For example, a new company (LifeStory Health) tests for modified proteins in menstrual blood for early-stage CVD detection, risk stratification, and treatments [47]. Wearable technologies such as wearable electrocardiogram devices (ECG) in a bra and smartwatches to monitor for cardiovascular disease and hypertension are also being developed [50]. Rigorous studies are needed to assess the actual use accuracy and predictive value of these technologies.

### 5.2. Obesity

In the United States, severe obesity (Body Mass Index (BMI) > 35 kg/m) is more prevalent in women than men. Sex-specific risk factors include monthly and lifetime hormone fluctuations, menopause, and pregnancy [48]. Specifically, E_2_ fluctuations lead to downregulation of metabolism and hunger-regulating hormones such as cholecystokinin (CCK), leptin, and peptide YY (PYY). Progesterone also increases cravings and energy intake [48]. Female-specific cloud-based health management systems, such as myAva, are filling this market to link health data (i.e., reproductive, vascular and metabolic health) and laboratory data with lifestyle recommendations and for health management. Nuanced data is also included, which includes proteomics, metabolomics, genomics, and microbiomics, used in part to form a Precision Nutrition Program which offers actionable intelligence.

### 5.3. Cancer

Precision Health for women is also applicable for cancer risk assessment and the early detection of cancer [51,52]. For this review, we will focus on women-specific cancers. At-home, self-collection tests are available that assess for the predisposition of genetic cancers [53,54]. For example, Color Genomics offers an at-home saliva test which uses a 30-gene panel to test for hereditary breast, ovarian, colorectal, melanoma, pancreatic, prostate, uterine and stomach cancers (APC, ATM, BAP1, CHEK2, MLH1, MSH2, MSH6, PMS2, POLE, PTEN, STK11, and TP53) [55]. While many of these genes have been detected in specific cancers [53,54], the functional risk of individual genetic alterations before detection of cancer is uncertain. The clinical and individual implications of the information available in over the counter genetic testing deserves a cautious approach; these options are quickly emerging in the commercial landscape and if integrated responsibly may play a role in targeted preventive screening.

#### 5.3.1. Cervical cancer

Cervical cancer is a malignant neoplasm of the uterine cervix, most commonly squamous cell carcinoma, but also includes endocervical adenocarcinoma. Human Papillomavirus is the causative agent of more than 99% of cervical squamous cell carcinomas, and is part of the screening algorithm [56]. Detection of hrHPV and/or precancerous lesions can prevent progression to invasive cancer. Current recommendations use a combination of screening via a cytology Pap smear and hrHPV testing [57,58,59]. Emerging technologies offer home testing for hrHPV in an effort to improve participation in and uptake of HPV screening [60,61]. Self-collect devices such as the Eve are being used in Canada, and similar such devices are being tested in the United States [60,61,62,63]. Integration of hrHPV home testing with follow-up care and/or Pap smear is a potential barrier to widespread impact. Therefore, testing with links to healthcare and counseling services will be needed to explain the implications of an infection and link women to clinical management.

#### 5.3.2. Breast cancer

Breast cancer is a malignant proliferation of cells within the breast tissue. The heterogeneity of the disease is apparent morphologically under the microscope, but also increasingly by hormone markers and molecular/gene expression [64]. Hormone receptors for estrogen receptor and progesterone receptor have been used in characterizing the cancer subtypes for decades. More recently, human epidermal growth factor receptor (HER2) testing is used to predict response to a drug targeted at the receptor (Herceptin) [65]. Treatments have been tailored to these receptors for decades [66]. Emerging classification based on multi-gene panels (e.g., OncotypeDx) predict the potential chemotherapy benefit in different risk groups [67,68]. Precision health options are now focusing on detection of breast cancer earlier. While biennial screening mammograms are recommended for women age 50–74 by the United States Preventive Services Task Force [69], only 60–70% of women pursue this screening [70]. Self-exam is not as reliable as the radiographic surveillance [69]. Some companies are now using adaptive mammography to design a bra that performs breast cancer screening (iSono Health) [71], often without the input of a healthcare provider. This technology is not yet available for clinical use, but if clinically proven, such technology may offer the potential to increase screening technology options and availability.

#### 5.3.3. Ovarian cancer

Ovarian cancer is a term that encompasses an array of benign to malignant entities. While high-grade ovarian cancer has the highest rate of mortality amongst gynecologic malignancies, early detection and distinction from benign mimics is lacking. Detection of ovarian cancers is notoriously difficult since physical symptoms often present late [72]. Furthermore, the most commonly used biomarker, CA125, is not particularly sensitive and can be elevated in benign conditions including endometriosis, pelvic trauma, ovarian cysts [72,73]. Scientists have been looking for innovative and ideally non-invasive ways to detect this cancer at earlier stages [72,74]. A group from Israel developed a breath sensor that uses molecularly modified gold nanoparticles that forms a unique nanopartical spatial configuration when they interact with specific volatile organic compounds. The device had 82% accuracy for detection of cancer irrespective of confounders such as tobacco and comorbidities [75]. Urine biomarkers are also being explored to detect ovarian carcinomas including human epididymis protein 4 (HE4) B-cell lymphoma 2 protein (Bcl-2), and have been proposed to discriminate between benign and malignant tumors [74]. The overall mortality and treatment response benefits have yet to be demonstrated, but utilization of urine is a more favorable body fluid in some respects due to ease of access, volume, and less complex protein matrix (as blood). While testing urine may hold promise in relatively noninvasive detection of disease, challenges still exist in overcoming environmental and hormonal influences that can alter the integrity of the fluid [74].

### 5.4. Mental Health

It is becoming increasingly apparent that sex-specific research and testing is important to provide individualized recommendations is needed to address mental health issues specific to women—Including postpartum depression, hormone balance and depression, and gender-specific depression management [2,76]. Twice as many women suffer from depression than men, and less than 1% of clinical trials analyze medications and outcomes by sex [77]. Women also have unique physiologic and hormonal changes that predispose them to depression during pregnancy, postpartum, and menopause. Given the larger burden of mental health issues among women, more resources are critical to address these needs. Online and virtual resources for mental health support are emerging, such as Maven which offers digital mental health resources. Mobile applications are also being developed for depression, stress, meditation, and eating disorder support. The University of North Carolina at Chapel Hill has developed a mobile application specifically for postpartum depression as part of a research study on the topic (PPD APP).

After the discovery that the effects of Gamma-amino butyric acid (GABA), and drugs associated with GABA, can be affected by sex hormones, there has been increasing awareness regarding the dosage and metabolism of medications in women [78]. Additional research about treatments that are most appropriate for women are still needed since very few of the anti-depressant clinical trials have analyzed based on sex [77]. Furthermore, pharmacogenetics studies may offer additional insights into appropriate dosage of medications used for mental health and sleep disorders based on sex [79,80].

### 5.5. Alzheimer’s Disease

Progressive dementia and cognitive impairment are hallmarks of Alzheimer’s disease (AD) [3] and two-thirds of people affected by this disease are women. Furthermore, while both men and women can carry the APOE Ɛ4 allele, which is the primary genetic risk factor for AD, women are more likely to experience a faster decline in memory and physical functioning when they carry the allele compared to men [3]. Tracking devices and mobile applications (e.g., AngelSense, GPS Smart Sole, iTraq) have emerged to provide caregivers with up-to-date information on the location of the affected patients and many integrate fall sensors and links to hotlines or emergency services. Additional research and technology aimed at understanding and supporting women with Alzheimer’s disease is therefore needed.

## 6. Conclusions

A plethora of new technologies are being developed for women in the FemTech space which have the potential to make an impact on improving women’s health. Limitations to the technology include the currently high device costs, limited data on the accuracy and unbiased clinical trial and actual use trials. In addition, regulatory and technical challenges will pose challenges to implementation and use. Nevertheless, these emerging devices and resources may provide a unique opportunity to provide access to education and testing especially among low-resource and marginalized populations who would otherwise be unable or unlikely to access care. There is potential to continue to link these technologies to health professionals and conventional medicine practice to improve health outcomes, especially for the unique health challenges facing women throughout their life.

## Figures and Tables

**Figure 1 jcm-08-01525-f001:**
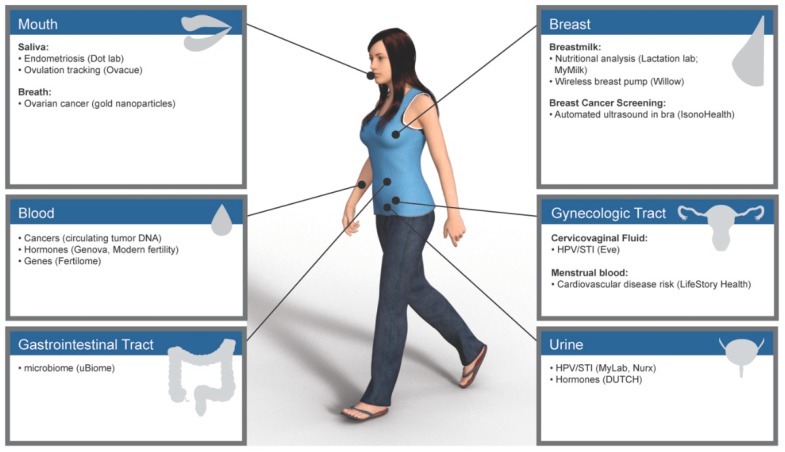
Emerging tests and technologies by anatomic site.

**Figure 2 jcm-08-01525-f002:**
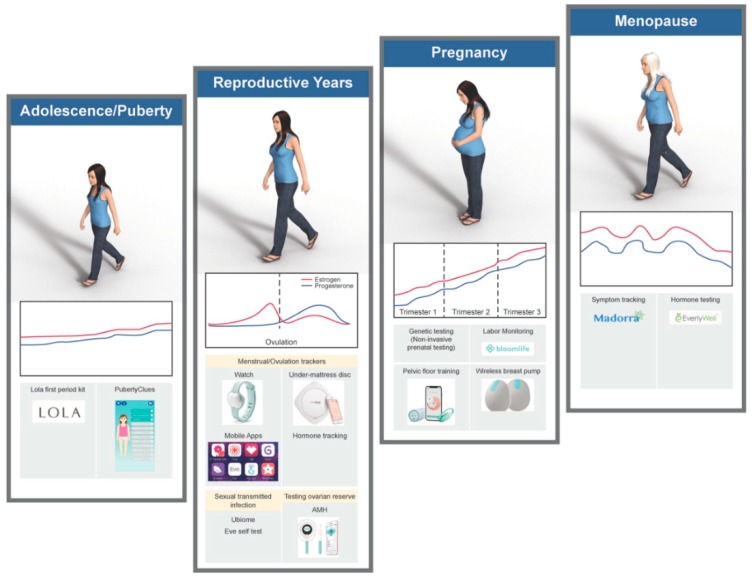
Lifestyle and hormone cycles with applicable technologies.

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
