# Peer review of "Advances in Precision Health and Emerging Diagnostics for Women"

_jcm, 2019, doi:10.3390/jcm8101525_

Round 1
Reviewer 1 Report
Line 25 - Correct spelling of Crone. The urbane dictionary definition of Chrone is "the space where the base of the shaft meets the top of the ballsack."
Line 32 - change to "has linked"
Line 49 - change to "prevention of disease"
Line 59 - lifetimes (add) during puberty
Line 64 -delete "hence"
Line 71 stimulate the
Line 77 delete "hence"
Line 126 delete furthermore
Line 233 correct spelling of Crone
Line 246 delete "infect"
Line 247 delete "futhermore"
Line 248 define E2
Line 267 define CCK and peptide YY
Line 270 for health management
Line 276 Capitalize "Color Genomics"
Liine 282 approach. Delete "however"
Line 329 Awkward sentence - is twice in the sentence
Line 332 change than to as
Author Response
Thank you very much for your detailed review of our manuscript. All of the updates requested have been made in the updated manuscript (attached here with mark-ups for review).

Reviewer 2 Report
The review article by Fitzpatrick and Thakor critically approaches several issues dealing with the recent advances in precision health and the emerging diagnostics for women. First of all, it is a very interesting topic for collecting information and integrating it in a well-organized article. Precision health offers the promise to provide personalized, female-specific lifestyle coaching, personal health education, and early disease detection, thus minimizing the consequences of diseases throughout a woman’s life. Contrarily, precision medicine focuses on molecular targeting to diagnose and treat diseases after the onset of a disease.
It is a well-written article, the questions are clear and authors have managed to give answers to most of them. Most important relevant articles have been analyzed and included in the literature. It is a reader-friendly article and figures are very comprehensive, thus contributing to the better understanding of the text. Figure 2, especially, referred to the association of lifecycle and hormone cycles with any applicable technologies looks superb.
Authors state (p.6) that chronic diseases-predictive tests, based on single nucleotide polymorphisms (SNPs) analyses, are still under development and it is correct. However, I would like to point out that this is a rather superficial task, given the contribution of transcriptomics (including microRNAs), epigenetics and environmental factors in the disease development apart from the genetic background (SNPs) mentioned.
Overall, it is an interesting article and the exploration of the emerging technologies, which can optimize the health of females through their lifecycle is complete, sufficient and clear.
Author Response
Thank you very much for the helpful feedback on our manuscript.
----
Your comment about the value of SNP analysis in chronic disease was particularly helpful, and we have added an additional explanation as well as citations to illustrate this concept:
The implications of detection of single nucleotide variants remain unclear given the contribution of epigenetics and environmental factors involved in disease development. For example, genetic association studies have identified single nucleotide variants associated with endometriosis, but these presumed etiological mutations are neither required for disease development nor detected in all cases of those with disease.
---
In addition, we have added additional references and discussion to the menopause section to improve appropriate and adequate references of the review.
These changes can be found in the updated manuscript attached.
---
Thank you again for your time and review of our article.
